# Electron cryo-microscopy structure of the canonical TRPC4 ion channel

**Deivanayagabarathy Vinayagam[1], Thomas Mager[2], Amir Apelbaum[1], Arne Bothe[1], Felipe Merino[1], Oliver Hofnagel[1], Christos Gatsogiannis[1], Stefan Raunser[1]***

[1]Department of Structural Biochemistry, Max Planck Institute of Molecular Physiology, Dortmund, Germany; [2]Department of Biophysical Chemistry, Max Planck Institute of Biophysics, Frankfurt am Main, Germany

**Abstract** Canonical transient receptor channels (TRPC) are non-selective cation channels. They are involved in receptor-operated $Ca^{2+}$ signaling and have been proposed to act as store-operated channels (SOC). Their malfunction is related to cardiomyopathies and their modulation by small molecules has been shown to be effective against renal cancer cells. The molecular mechanism underlying the complex activation and regulation is poorly understood. Here, we report the electron cryo-microscopy structure of zebrafish TRPC4 in its unliganded (apo), closed state at an overall resolution of 3.6 Å. The structure reveals the molecular architecture of the cation conducting pore, including the selectivity filter and lower gate. The cytoplasmic domain contains two key hubs that have been shown to interact with modulating proteins. Structural comparisons with other TRP channels give novel insights into the general architecture and domain organization of this superfamily of channels and help to understand their function and pharmacology.
DOI: https://doi.org/10.7554/eLife.36615.001

**\*For correspondence:**
stefan.raunser@mpi-dortmund.mpg.de

**Competing interests:** The authors declare that no competing interests exist.

## Introduction

Transient receptor potential (TRP) channels constitute a large superfamily of ion channels that can be grouped in seven subfamilies: TRPC, TRPM, TRPML, TRPP, TRPV, TRPA and TRPN (*Montell, 2005*). The channels of the TRP superfamily are diverse in respect to modes of activities, ion selectivity and physiological functions. Most TRP channels are non-selective cation channels with varying preferences for $Ca^{2+}$ over $Na^+$ (*Owsianik et al., 2006*). Some TRPs are physically stimulated or voltage-activated, whereas others respond to the binding of ligands or the direct interaction with other proteins. Corresponding to their functional diversity TRPs are involved in many cellular processes, including mechanosensation, thermosensitivity, nociception and store-operated $Ca^{2+}$ entry (*Nilius and Flockerzi, 2014*).

TRPC4 is one of seven members of the subfamily of TRPC (canonical TRPs) channels (*Freichel et al., 2014*). TRPC channels are $Ca^{2+}/Na^+$-permeable cation channels that are expressed in many cell types and tissues, including brain, placenta, adrenal gland, retina endothelia, testis, and kidney (*Freichel et al., 2014*). The channels play an important role in vasorelaxation and neurotransmitter release. TRPC4 contributes to axonal regeneration after nerve injury (*Wu et al., 2008*) and TRPC channels in general are necessary mediators of pathologic cardiac hypertrophy (*Wu et al., 2010*).

The activation mechanism of TRPC4 has been controversially discussed (*Plant and Schaefer, 2003*). Dependent on the cellular environment and method of measurement, the reported activation mechanism, permeability and biophysical properties differ for TRPC4 and its close homologue TRPC5 (*Plant and Schaefer, 2003*). Several studies showed that TRPC4 and TRPC5 form receptor-operated, $Ca^{2+}$-permeable, non-selective cation channels (*Okada et al., 1998*; *Schaefer et al.,*

*2000*). Others found that TRPC4 and TRPC5 are activated by $Ca^{2+}$ store-depletion with moderate to high $Ca^{2+}$ permeabilities (*Philipp et al., 1996*; *Warnat et al., 1999*). In line with both findings, TRPC4 directly interacts with $IP_3$ receptors, calmodulin (*Mery et al., 2001*; *Tang et al., 2001*), STIM1 (*Lee et al., 2010*; *Zeng et al., 2008*), the lipid-binding protein SESTD1 (*Miehe et al., 2010*) and the G protein $G_{\alpha i2}$ (*Jeon et al., 2012*). SESTD1 binds several phospholipid species and is essential for efficient receptor-mediated activation of TRPC5 (*Miehe et al., 2010*). In addition, TRPC channels have been shown to be activated by NO (*Yoshida et al., 2006*).

Recently, (-)-Englerin A has been shown to be a potent and selective activator of TRPC4 and TRPC5 calcium channels (*Akbulut et al., 2015*). It selectively kills renal cancer cells by elevated $Ca^{2+}$ influx. (-)-Englerin A is so far the only known potent activator of TRPC4 and TRPC5.

The resolution revolution in cryo-EM (*Kuhlbrandt, 2014*) brought an enormous amount of high-resolution structures of TRP channels in the last 5 years (*Madej and Ziegler, 2018*). The cryo-EM structure of TRPV1 (*Liao et al., 2013*) ushered structural biology of TRP channels in a new era. So far structure models for 48 TRP channels from six subfamilies have been published (*Madej and Ziegler, 2018*). The TRPCs represent the only subfamily for which no high-resolution structure has been reported so far limiting our understanding of these important types of cation channels. Here, we present the first cryo-EM structure of zebrafish TRPC4 in its unliganded (apo), closed state.

## Results and discussion

Initially, we planned to heterologously express human TRPC4 in HEK293 cells and purify it to determine its structure by single particle cryo-EM. However, the protein proved not to be stable enough for structural investigations. We therefore screened TRPC4 orthologues from several different species and found wild type TRPC4 from zebrafish ($TRPC4_{DR}$) to be the most suitable for our studies. $TRPC4_{DR}$ has very high sequence similarity to human TRPC4 (76% sequence identity, *Figure 1—figure supplement 1*).

To determine whether $TRPC4_{DR}$ has the same channel properties as human TRPC4, we performed voltage-clamp experiments with HEK293 cells heterologously expressing $TRPC4_{DR}$. The measurements demonstrated that, like human TRPC4 (*Akbulut et al., 2015*), $TRPC4_{DR}$ can be activated by (-)-Englerin A resulting in similar currents (*Figure 1*). Of note, current-voltage curves (IV-curves) in the presence of (-)-Englerin A showed a doubly rectifying form (*Figure 1e*) and reversal potentials close to 0 mV ($-3.2 \pm 1.7$ mV, n = 6). The doubly rectifying form of the IV-curve is a characteristic hallmark of active TRPC4 (*Akbulut et al., 2015*; *Freichel et al., 2014*; *Schaefer et al., 2000*). Under the experimental conditions, namely asymmetric ion concentrations, reversal potentials close to 0 mV indicate a poor cation selectivity, which is a known property of several TRPC4 variants from different species (*Freichel et al., 2014*; *Schaefer et al., 2002*).

We overexpressed Strep-tagged $TRPC4_{DR}$ in HEK293, solubilized it in n-dodecyl-β-d-maltopyranoside (DDM)/cholesteryl hemisuccinate (CHS) and purified it using affinity and size exclusion chromatography (*Figure 2—figure supplement 1a–b*). After purification the detergent was exchanged against amphipols. The resulting sample was homogeneous and suitable for structural studies (see Materials and methods, *Figure 2—figure supplement 1c–e*). We then solved the structure of $TRPC4_{DR}$ in its unliganded (apo), closed state by cryo-EM and single-particle analysis (see Materials and methods) at an average resolution of 3.6 Å (*Figure 2a*, *Table 1*, *Figure 2—figure supplements 2–4*, *Video 1*). The high quality of the map allowed us to build a model of 70% of $TRPC4_{DR}$ de novo. The final model contains residues 18–753 with some loops missing (*Figure 2b*). As in most other TRP structures the C-terminal region (residues 754–915) could not be resolved indicating that this region is highly flexible.

The overall structure of the homotetrameric $TRPC4_{DR}$ is similar to that of other TRP channels (*Figure 2a–b*). Especially the transmembrane domain comprising the voltage-sensor-like (VSL) domain and pore domain is structurally conserved. Like TRPVs, TRPMs, TRPAs and TRPNs, $TRPC4_{DR}$ does not have extracellular but extended cytoplasmic domains. The resolved regions of the cytoplasmic domain of $TRPC4_{DR}$ reaches ~80 Å into the cytosol and is relatively short compared to other TRP channels. The cytoplasmic domain can be separated in an upper and lower part. The upper part comprises a conserved TRP domain, a pre-S1 elbow domain that enters partially the membrane and an extended helical linker domain (*Figure 2c–d*). The lower part is formed by the Rib (or Stretcher) helix, the C-terminal helix and four ankyrin repeats (*Figure 2c–d*). 3-D classification and refinement

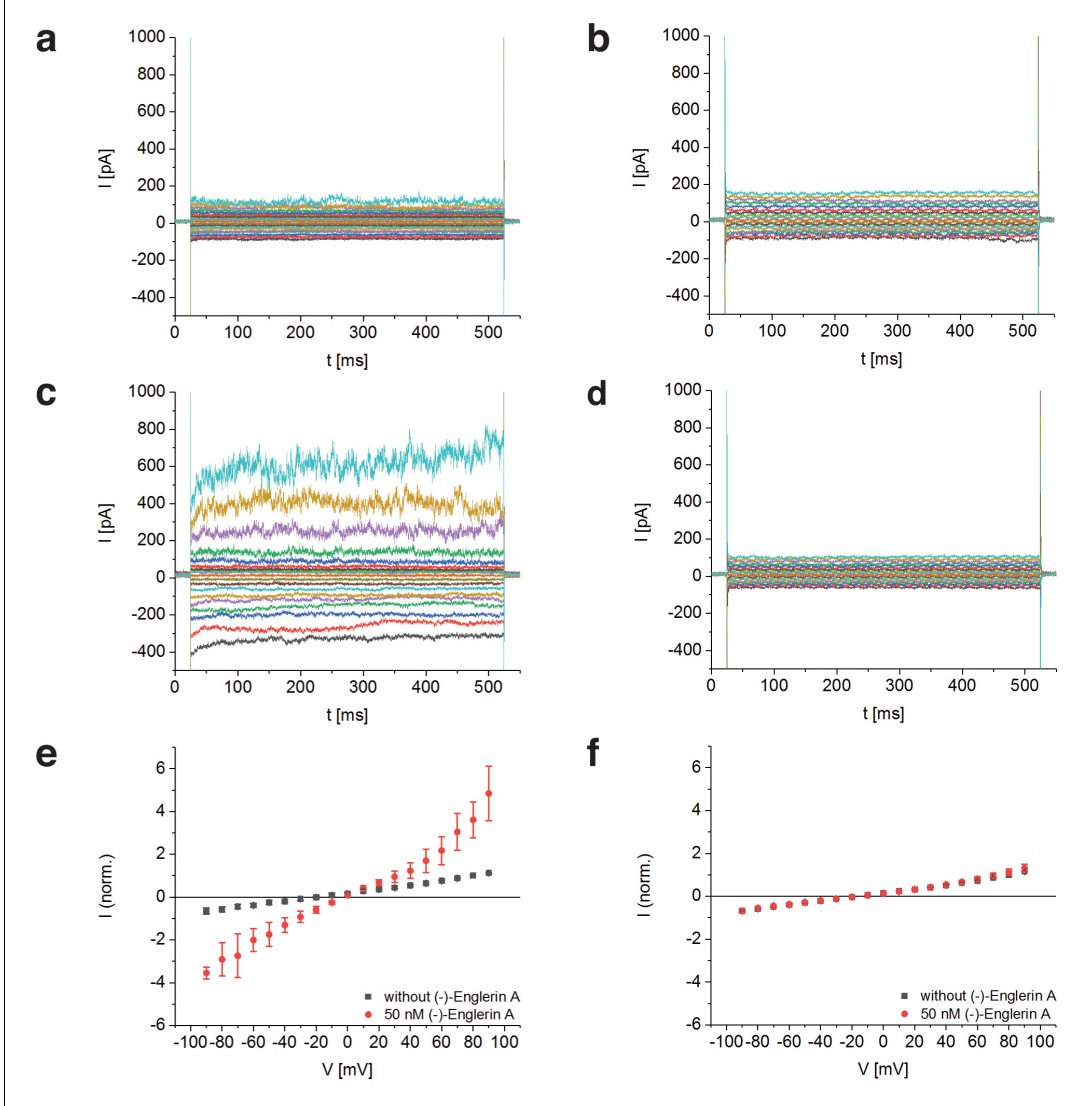

**Figure 1.** Activation of TRPC4$_{DR}$ by the selective activator (-)-Englerin A. (a–f) HEK293 cells heterologously expressing TRPC4$_{DR}$-EGFP (a,c,e) and untransfected control cells (b,d,f) were investigated by voltage-clamp experiments in the whole-cell configuration. The membrane potentials were clamped to values ranging from −90 to +90 mV in the absence (a,b) and in the presence (c,d) of 50 nM (-)-Englerin A. Upon addition of 50 nM of (-)-Englerin A, the current density at −60 mV increased from −3.1 ± 1.9 pA/pF (n = 6) to −16.7 ± 10.7 pA/pF (n = 6). In untransfected control cells, the current density in the absence and presence of the activator was virtually the same with values of −2.1 ± 1.2 pA/pF (V = - 60 mV, n = 5) and −1.8 ± 0.9 pA/pF (V = −60 mV, n = 5) respectively. (e,f) Current-voltage curves in the absence (black squares) and in the presence (red circles) of 50 nM (-)-Englerin A. Currents were normalized to the current value in the absence of (-)-Englerin A at a membrane potential of +80 mV. Note that the measurements in the absence and in the presence of (-)-Englerin A were performed on the same cells. Shown are the normalized mean currents of 6 (e) and 5 (f) different cells. Error bars are ± SEM. The measurements were performed as described in Materials and methods.

DOI: https://doi.org/10.7554/eLife.36615.002

The following figure supplement is available for figure 1:

**Figure supplement 1.** Multiple sequence alignment of TRPC4$_{DR}$, human TRPC4 and human TRPC5.

DOI: https://doi.org/10.7554/eLife.36615.003

of the final data set (*Figure 2—figure supplement 3*) revealed that the lower cytoplasmic part of TRPC4$_{DR}$ is flexible, whereas the transmembrane domain is not (*Video 2*).

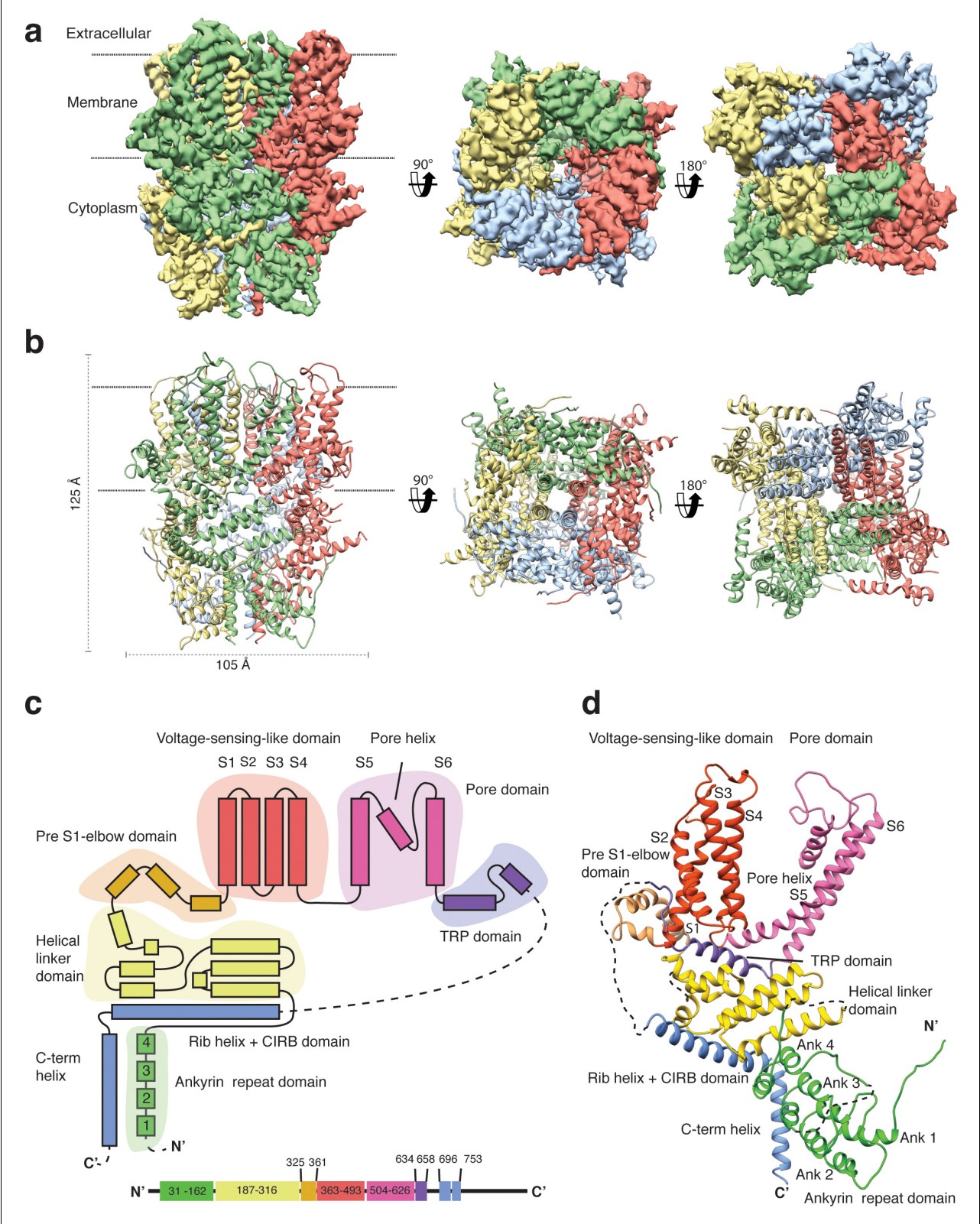

**Figure 2.** Structure of TRPC4$_{DR}$. (a) Cryo-EM density map of TRPC4$_{DR}$ with each protomer colored differently and shown as side, bottom and bottom view. (b) Ribbon representation of the atomic model of TRPC4$_{DR}$. Colors are the same as in (a). (c) Topology diagram depicting the domain organization of a TRPC4$_{DR}$ protomer. (d) Ribbon representation of a TRPC4$_{DR}$ protomer. Each domain is shown in a different color and labeled accordingly.

*Figure 2 continued on next page*

*Figure 2 continued*

DOI: https://doi.org/10.7554/eLife.36615.004

The following figure supplements are available for figure 2:

**Figure supplement 1.** Purification of TRPC4$_{DR}$ and negative stain EM of TRPC4$_{DR}$ in amphipols.
DOI: https://doi.org/10.7554/eLife.36615.005
**Figure supplement 2.** Cryo-EM structure of TRPC4$_{DR}$.
DOI: https://doi.org/10.7554/eLife.36615.006
**Figure supplement 3.** Single particle processing workflow for TRPC4$_{DR}$ structure determination.
DOI: https://doi.org/10.7554/eLife.36615.007
**Figure supplement 4.** Comparison of the $Ca^{2+}$ binding site in TRPM4 and TRPC4$_{DR}$.
DOI: https://doi.org/10.7554/eLife.36615.008
**Figure supplement 5.** LFW motif of the pore region and cysteines involved in disulphide bridges in the TRPC4$_{DR}$ structure.
DOI: https://doi.org/10.7554/eLife.36615.009

## Voltage-sensor like domain and ion conducting pore

Although TRP channels contain a structurally conserved VSL domain comprising four helices (S1-S4), only some members of the TRP family, such as TRPM8, TRPM4, and TRPP1 have been shown to exhibit mild voltage sensitivity (*Nilius et al., 2003*; *Shen et al., 2016a*; *Voets et al., 2007*). Helix S4

**Table 1.** EM data collection and refinement statistics of TRPC4$_{DR}$

| Data collection | Data set 1 [Data set 2] |
| --- | --- |
| Microscope | Titan Krios (Cs corrected, XFEG) |
| Voltage (kV) | 300 |
| Camera | K2 summit (Gatan) |
| Pixel size (Å) | 1.09 [1.09] |
| Number of frames | 60 [40] |
| Total electron dose (e⁻/Å²) | 69 [74] |
| Number of particles | 132,622 |
| Estimated defocus range | 0.844–2.931 |
| Atomic model composition | |
| Non-hydrogen atoms | 21,412 |
| Protein atoms | 21,192 |
| Ligand atoms | 236 |
| Refinement (Phenix) | |
| RMSD bond | 0.01 |
| RMSD angle | 1.11 |
| Model to map fit, CC mask | 0.80 |
| Resolution (FSC@0.143, Å) | 3.6 |
| Map sharpening B-factor (Å²) | −130.0 |
| Validation | |
| Clashscore | 3.53 |
| Poor rotamers (%) | 0.17 |
| Favoured rotamers (%) | 94.48 |
| Ramachandran outliers (%) | 0.0 |
| Ramachandran favoured (%) | 92.38 |
| Molprobity score | 1.62 |
| EMRinger score | 2.42 |

DOI: https://doi.org/10.7554/eLife.36615.010

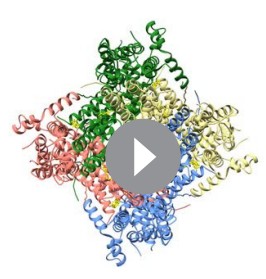

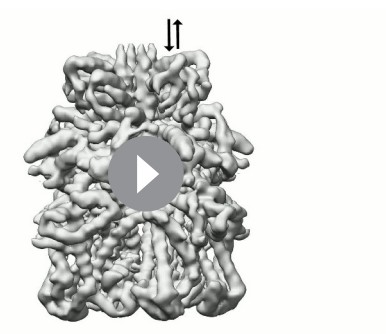

**Video 1.** Overview of the TRPC4$_{DR}$ structure
DOI: https://doi.org/10.7554/eLife.36615.011

**Video 2.** Morph between conformational states of TRPC4$_{DR}$
DOI: https://doi.org/10.7554/eLife.36615.012

harbors the key residues involved in voltage sensing in voltage-gated ion channels (*Swartz, 2008*). A general motif in the voltage-sensing domain (VSD) of voltage-gated ion channels comprises two sets of basic residues on helix S4 that are stabilized by counteracting acidic residues and separated by a hydrophobic patch (*Kim and Nimigean, 2016*). Upon voltage change, helix S4 moves and the positively charged residues jump from one counteracting set of residues over the hydrophobic patch to the next set of negatively charged residues (*Zhang et al., 2012*).

Comparing the VSL domain in our structure and other TRP channels with the VSD in the structure of the Kv1.2-Kv2.1 paddle chimera (*Long et al., 2007*), we found that R491 in TRPC4$_{DR}$ is located at the same position as K302 in the potassium channel, corresponding to the lower pair of electrostatically interacting residues (*Figure 3a*). At this position an arginine has been observed in TRPM4 and TRPM8 (*Winkler et al., 2017*; *Yin et al., 2018*), and a lysine in TRPP1 (*Shen et al., 2016b*) (*Figure 3*). In all cases, the basic residues are stabilized by interaction with acidic residues. Importantly, there are no equivalently interacting residues in TRPV1 at this position (*Liao et al., 2013*) (*Figure 3a*). The hydrophobic patch that has been shown to maintain the electric field across the membrane by preventing the movement of ions (*Lacroix et al., 2014*) is more prominent in all compared TRP channels than in the Kv1.2-Kv2.1 paddle chimera (*Figure 3a*). However, the second pair of interacting basic and acidic residues, which is essential for the movement of helix S4 comprising residues R290, R293, E183 and E226 in Kv1.2, cannot be found in the TRP channels. Only TRPM8 has two residues with complementary charges at this position, however, these residues point away from each other in the structure and do not interact (*Yin et al., 2018*) (*Figure 3a*). Thus, the mechanism of voltage gating in TRP channels must differ from that of voltage-gated ion channels.

Interestingly, positively charged residues that are involved in voltage sensing in Kv1.2, such as R293, R296, R299, R302, R305, and K308 are replaced by polar residues, including N485, S488, S494, and T497 in TRPC4$_{DR}$. The VSL domain in other TRP channels contains more hydrophobic residues (*Liao et al., 2013*; *Winkler et al., 2017*) (*Figure 3b*).

Upstream of the VSL domain resides the pre-S1 elbow domain. It sits inside the membrane and forms a cavity with helices S1 and S4 in which we could identify density corresponding to a sterol (*Figure 4a–b*). Since we added CHS during the purification of TRPC4$_{DR}$, we fitted this molecule into the density. Interestingly, a CHS molecule has also been found at the same position in the structures of TRPM4 (*Autzen et al., 2018*) and TRPML3 (*Hirschi et al., 2017*). In a second cavity between the pre-S1 elbow domain and helices S1 and S2 resides a proline-rich repeat connecting the TRP and helical linker domains (see below).

The activation of TRPC4 and TRPC5 is dependent on Ca$^{2+}$ (*Plant and Schaefer, 2003*). However, so far no Ca$^{2+}$ binding sites have been described for TRPCs and we do not have Ca$^{2+}$ in our sample buffer. Interestingly, the four coordinating residues that have been reported for binding of Ca$^{2+}$ in Ca$^{2+}$-activated TRPM4 (*Autzen et al., 2018*) are conserved in TRPC4 (*Figure 2—figure supplement 4*). In the TRPC4$_{DR}$ structure, their position is very similar to TRPM4 and would allow the coordination of a Ca$^{2+}$ ion (*Figure 2—figure supplement 4*). Although we indeed observed an extra density

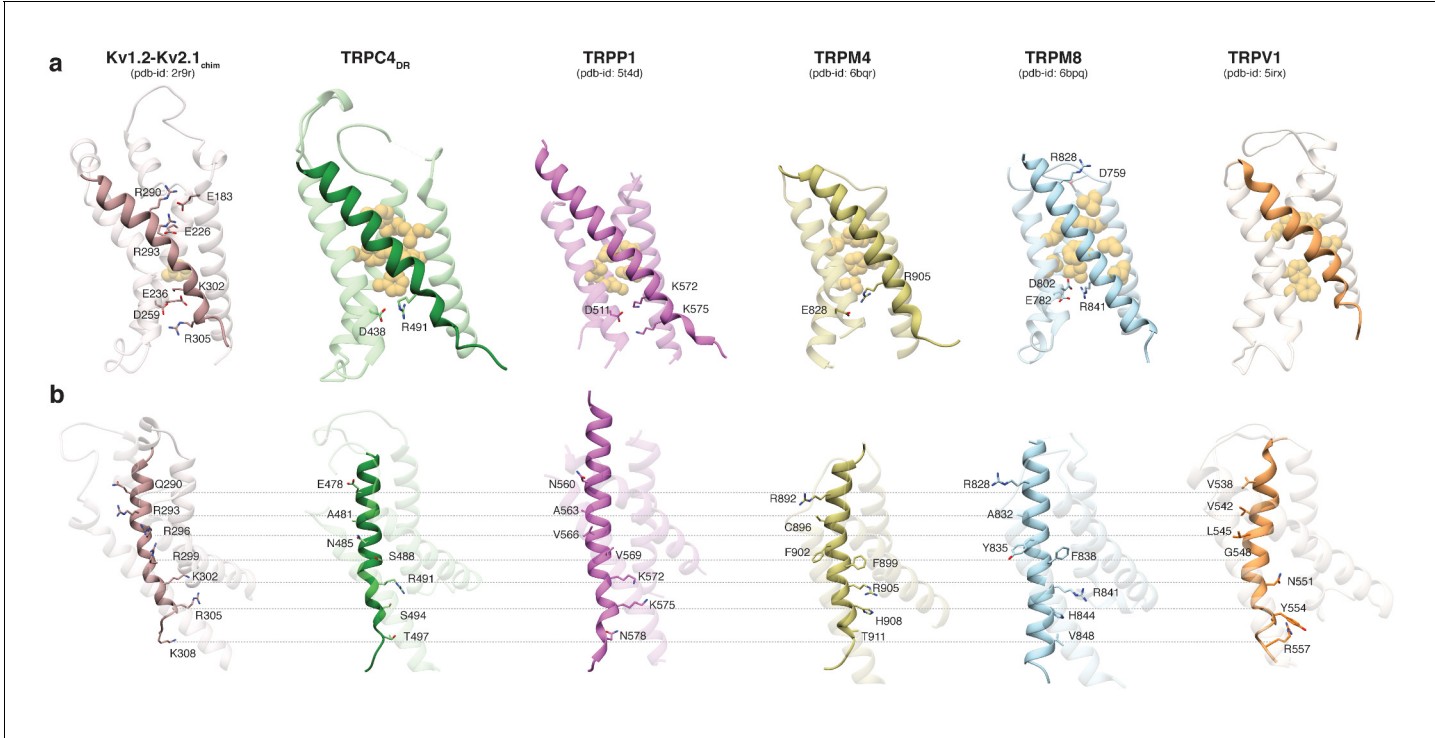

**Figure 3.** Comparison of the VSL domain of selected TRP family members with the voltage-sensing domain of the chimeric $K_V$1.2-$K_V$2.1 channel. (**a**) The voltage-sensing domain of the chimeric $K_V$1.2-$K_V$2.1 channel and the VSL domains of $TRPC4_{DR}$, TRPP1, TRPM4, TRPM8 and TRPV1 are shown in ribbon representation. The S4 helix in each case is highlighted with dark shaded color. The rest of the domains are shown in light transparent color. The residues which form the hydrophobic patch in the middle of the domain are shown in golden yellow sphere representation with light transparency. The residues involved in ion-pair interactions are shown in stick representation. (**b**) The structures in (**a**) are rotated to better view the residues important for voltage sensing in the S4 helix of the chimeric $K_V$1.2-$K_V$2.1 channel and topological equivalent residues of TRP channels are shown in stick representation and labeled.

DOI: https://doi.org/10.7554/eLife.36615.013

at the center between these residues, we did not assign it to $Ca^{2+}$ because of the limited resolution (*Figure 2—figure supplement 4d*).

Like in other TRP channels, helix S5 and S6 swap over to the respective helices of the adjacent protomer and form the pore at the center of the tetrameric channel (*Figure 2b*). The extracellular opening of the pore is negatively charged (*Figure 5a*) and the residue E555 at the tip of the pore turret, which is conserved in TRPC1, TRPC4, and TRPC5 (*Liu et al., 2003*), forms a salt bridge with R556 of the adjacent protomer (*Figure 2—figure supplement 5a–b*). This results in a positioning of E555 away from the pore and in a relatively wide opening of its entrance similar to TRPV1 (*Liao et al., 2013*) (*Figure 5b–d*). This is in contrast to the narrow pore openings of the more selective $Ca^{2+}$ channels TRPV5 (*Hughes et al., 2018*) and TRPV6 (*McGoldrick et al., 2018*) and possibly explains the lower selectivity of TRPC4 channels. Mutation of E555 (E559 in TRPC1) to lysine results in decreased store-operated $Ca^{2+}$ influx (*Liu et al., 2003*), suggesting that a negatively charged and stable pore opening is crucial for the undisturbed permeation of cations through TRPCs.

The selectivity filter is formed by four glycine residues (G577) that constrict the pore to a diameter of 7.0 Å (3.7 Å defined by opposing van der Waals surfaces) (*Figure 5c–d*). Since the diameters of $Na^+$ and $Ca^{2+}$ ions are ~2 Å in their dehydrated and ~10–12 Å in their fully hydrated state, the cations are likely partly dehydrated when passing through the selectivity filter. Although TRPs differ in their cation selectivity, there is no clear factor recognizable that determines the level of selectivity (*Table 2*). Neither the diameter of the filter, ranging from 1.8 to 8.4 Å, nor the type of residue, mostly glycine or aspartate, correlates with the selectivity of the channels for $Na^+$ or $Ca^{2+}$ (*Table 2*). The level of selectivity must therefore be defined differently.

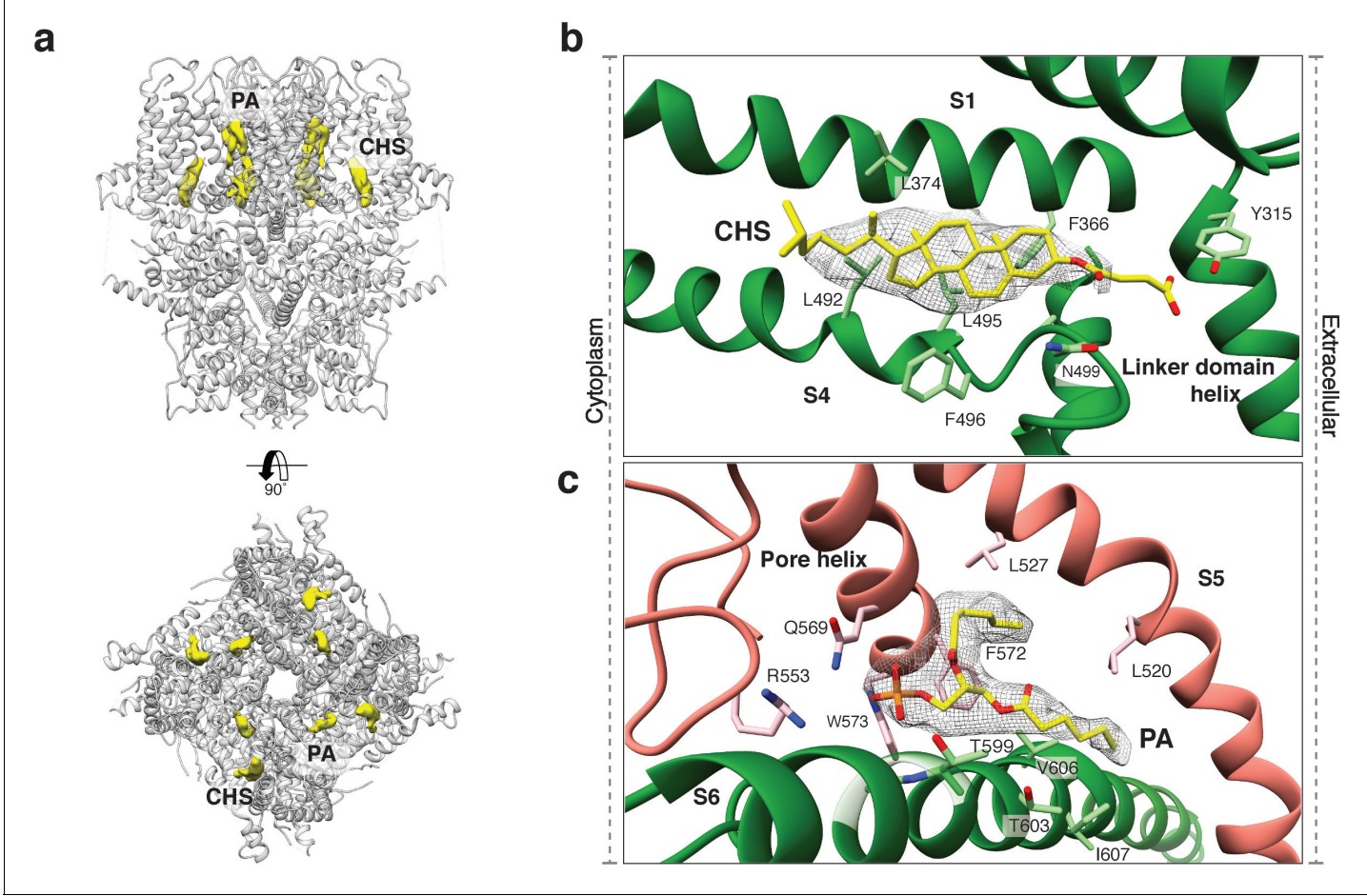

**Figure 4.** Lipid binding sites in the TRPC4DR structure. (**a**) Side and top view of the TRPC4DR structure with lipid densities highlighted in yellow against the model shown in ribbon representation. (**b–c**) Zoomed-in view on the cholesteryl hemisuccinate (CHS) and phosphatidic acid lipid (PA) binding sites, respectively. CHS and PA molecules are shown in yellow stick representation.

DOI: https://doi.org/10.7554/eLife.36615.014

Directly above the selectivity filter resides S580 (*Figure 5c–d*) which is replaced by an asparagine residue in human TRPC4 and TRPC5 (*Figure 1—figure supplement 1*). In TRPC5 this residue has been shown to be important for Ca$^{2+}$ permeability (*Chen et al., 2017b*) and in TRPV1 D646, located at a similar position, has been identified as important residue for cation selection (*García-Martínez et al., 2000*).

The lumen of the pore below the selectivity filter is mainly hydrophobic (*Figure 5b*) and leads to the lower gate at the cytoplasmic end formed by the conserved residues I617 and N621 (*Figure 5c*). I617 belongs to the M3 motif found in all TRPCs, TRPVs, and TRPMs (*Freichel et al., 2014*). In our case, the TRPC4DR channel is almost completely closed. The four isoleucines and four asparagines constrict the pore to a diameter of ~5.3 and ~4.2 Å, respectively (1.6 and 0.7 Å defined by opposing van der Waals surfaces) (*Figure 5c–d*). Ca$^{2+}$ and Na$^{+}$ ions are too large to fit through these constrictions even in a dehydrated state. The lower constriction of the TRPP channel PKD2 is also formed by an asparagine and leads to a closure of the pore (*Shen et al., 2016b*). Below the asparagine and very close to the cytoplasmic end of the pore, a glutamine (Q625) points towards the pore (*Figure 5c*). This residue is conserved in all TRPCs (*Figure 1—figure supplement 1*), however, its significance has remained unknown.

An important region inside the pore domain is the LFW motif (residues 571–573), which is also conserved in TRPC1 and TRPC5. In TRPC5, mutation of the LFW motif to AAA results in non-functional but folded channels (*Strübing et al., 2003*). In our TrpC4DR structure F572 and W573 form

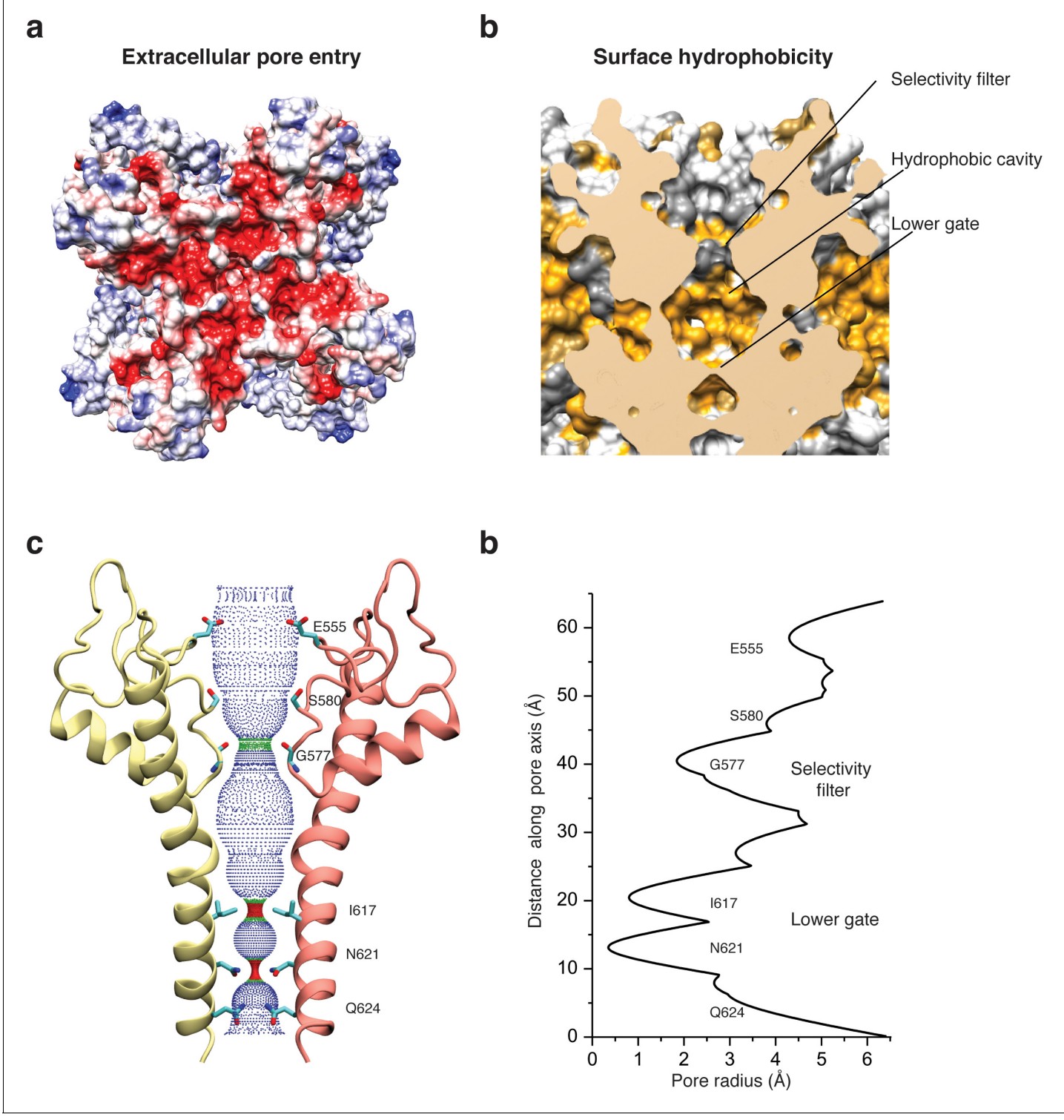

**Figure 5.** Architecture of the pore domain. (**a**) Surface electrostatic Coulomb potential at the extracellular mouth of TRPC4$_{DR}$. (**b**) Hydrophobic surface of the pore shown in vertical cross section. Hydrophobic patches are colored orange. (**c**) Ion conduction pore of TRPC4$_{DR}$ shown with diagonally facing protomers shown in ribbon representation. Critical residues important for gating and selection are shown in stick representation. (**d**) Pore radius determined along the pore axis using HOLE (**Smart et al., 1996**).

DOI: https://doi.org/10.7554/eLife.36615.015

**Table 2.** Comparison of selectivity filters among TRP family members

| TRP family name | Residue | Van der waals diameter (Å) | Selectivity* PCa: PNa | Reference |
|---|---|---|---|---|
| TRPC4$_{DR}$ | G577 | 3.7 | 7 | |
| TRPV1 | G643 | 1.5 | 3.8–9.6 | (*Liao et al., 2013*) |
| TRPV2 | Gly604 (M606) | 1.9 (1.35) | 3 | (*Zubcevic et al., 2016*) |
| TRPML3 | G457 (D459) | 2.0 (4.2) | Highly selective for Ca$^{2+}$ | (*Zhou et al., 2017*) |
| TRPML1 | G470 (D471) | 1.4 (2.1) | N.D. | (*Schmiege et al., 2017*) |
| TRPN (NOMPC) | G1506 | 3.8 | N.D. | (*Jin et al., 2017*) |
| TRPA1 | D915 | 3.2 | 0.8 | (*Paulsen et al., 2015*) |
| TRPV5 | D542 | 2.6 | >100 | (*Hughes et al., 2018*) |
| TRPV6 | D541 | 0.9 | >100 | (*Saotome et al., 2016*) |
| TRPM4 | G972 | 4.3 | PNa: PCa > 100 | (*Guo et al., 2017a*) |
| TRPP1 (PKD2) | Leu641 (Gly642) | 1.7 (3.6) | PNa: PCa > 100 | (*Shen et al., 2016a*) |

*(*Venkatachalam and Montell, 2007*)

DOI: https://doi.org/10.7554/eLife.36615.016

part of a prominent hydrophobic contact between the pore helix of one protomer and helix S6 of the adjacent protomer, stabilizing the pore (*Figure 2—figure supplement 5c–d*). The triple AAA mutation in this region likely destabilizes this interaction and results in the collapse of the pore explaining the inactivity of the channel. Interestingly, we found an additional density corresponding to an annular lipid at this interface (*Figure 4a,c*). The shape of the density clearly corresponds to phosphatidic acid (PA) or ceramide-1 phosphate. Since PA has been assigned to a similar density found in polycytin-2 (*Wilkes et al., 2017*) and PA has been included in our purification buffer we fitted PA into the density (*Figure 4c*). The phosphate head group interacts with the peptide backbone nitrogen and the polar side chains of T599 of helix S6 and the side chain amino and amide groups of W573 and Q569 of the pore helix, respectively (*Figure 4c*). The non-polar tail group is stabilized by interaction with multiple hydrophobic residues from the surrounding helices S5, S6 and the pore helix (*Figure 4c*).

Another well-studied region of the pore domain contains two cysteines (C549 and C554). Yoshida et al. showed that S-nitrosylation of these residues in TRPC5 leads to the activation of the channel (*Yoshida et al., 2006*). The authors proposed that the modification of the residues has a direct effect on the conformation of helix S6, resulting in the opening of the gate. In the TRPC4 structure, however, C549 and C554 do not locate in close proximity to helix S6 (*Figure 2—figure supplement 5a–b*). We can therefore exclude a direct interaction. The cysteines form a disulfide bridge in TRPC4$_{DR}$ (*Figure 2—figure supplement 5a–b*) and are likely involved in putting E555 in place, which is located at the extracellular vestibule above the selectivity filter (see above). Nitrosylation, that reduces the disulphide bond of cystines (*Yoshida et al., 2006*) could thus lead to a destabilization of the upper region of the selectivity filter. In this process, E555 could be released and orient towards the center of the pore, altering its properties. Thus, the conformational change could result in a negatively charged sink at the turret, thereby attracting cations, possibly explaining the effect of the increased Ca$^{2+}$ conductance observed by Yoshida et al. (*Yoshida et al., 2006*).

## The TRP domain and helical linker domain

The TRP domain and linker domain connect the transmembrane domain with the lower cytoplasmic domain (*Figure 2c–d*). The TRP domain (residues E634 – N658) resides directly after helix S6 and is sandwiched between the linker and transmembrane domains. It is conserved in all TRP subfamilies except TRPPs and TRPMLs (*Madej and Ziegler, 2018*; *Venkatachalam and Montell, 2007*). The core of the TRP domain consists of the conserved WKXQR TRP box sequence (residues 635–639). Its

central residue W635 has been shown to be of crucial importance for gating of several TRP channels. Its mutation in TRPV3 leads to Olmsted syndrome (*Lin et al., 2012*) and in NOMPC an exchange to alanine results in a channel with a prominent basal current that does not respond to mechanical stimuli anymore (*Jin et al., 2017*).

Like in other TRP domain-containing TRPs, W635 in TRPC4$_{DR}$ interacts with the conserved G503 in the adjacent M2 motif (residues 502–511) thereby linking the TRP domain with the linker between helices S4-S5 (*Figure 6a–b*). G503 is conserved in all TRP subfamilies containing a TRP domain (*Figure 6c*). Mutation of this residue to serine in TRPC4 and TRPC5 forces the channels in an open conformation (*Beck et al., 2013*). The proper interaction between G503 and W635 guarantees the stabilization of helix S6 that forms the lower gate of the pore. It becomes clear from looking at these residues in our TRPC4$_{DR}$ structure, that a G503S or W635A mutation would impair the interaction at this site and result in the loss of control over the gate (*Figure 6b*).

E648 and E649 which are conserved in TRPC4 and TRPC5 (*Figure 1—figure supplement 1*) are located at a peripheral loop of the TRP domain (*Figure 7a*). The glutamates interact electrostatically with two lysine residues (K684 and K685) in Stim1 proteins resulting in the activation of cation currents (*Lee et al., 2010*; *Zeng et al., 2008*). In line with this biochemical finding, E648 and E649 (E649 is not resolved in our structure) are exposed at the periphery of TRPC4$_{DR}$ (*Figure 7a*) and are accessible for interactions with other proteins.

TRPC4$_{DR}$ contains two conserved proline-rich regions downstream of the TRP domain. Proline-rich regions are often involved in specific protein–protein interactions and signal transmission. In TRPC4$_{DR}$ the first proline-rich region formed by P654, P655, P656 re-enters the membrane and is sandwiched between the pre-S1 elbow domain and helix S2 making it inaccessible from the cytoplasm or the membrane. The second proline-rich region around the residues P661 and P663 is not resolved in our structure but this region is located at the outside of the protein and could thereby be accessible for interacting with regulatory proteins like Homer. Homer is an adaptor protein that facilitates the physical interaction between TRPC1 and the IP$_3$ receptor (*Yuan et al., 2003*).

The helical linker domain is made of three layers of helices that are arranged like stair steps and a coil of shorter helices (*Figure 2d*). The upper layer of helices harbors the M1 motif (residues 285–319), which is conserved in the TRPC family (*Flockerzi, 2007*). The central two helices form a short coiled-coil structure that harbors the multimerization motif (residues D254 -Q302), that has been predicted to be involved in the multimerization of TRPCs (*Freichel et al., 2014*). Indeed, in contrast to other TRP family members (*Madej and Ziegler, 2018*), the helical linker domain does not only connect the transmembrane domain with the lower cytoplasmic domain but also strongly interacts with the adjacent linker domains thereby stabilizing the tetramer (*Figure 2a–b*). This kind of

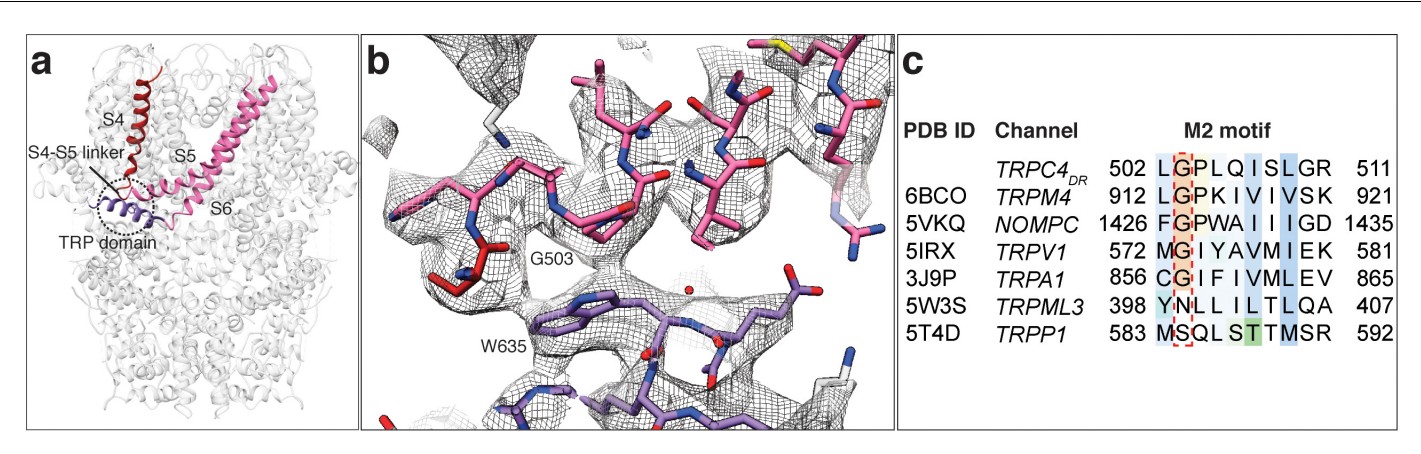

**Figure 6.** Interaction of the TRP domain with a conserved glycine in the S4-S5 linker. (a) Ribbon overview indicating the interaction site between the TRP domain and the S4-S5 linker. Key helices are shown in different colors. The rest of the protein is shown in light grey. (b) Zoomed-in view on the interaction site with key residues labeled. The chain trace is shown in stick representation. (c) Structure-based sequence alignment of TRP family members. The conserved glycine in the M2 motif is highlighted by a red dotted box.
DOI: https://doi.org/10.7554/eLife.36615.017

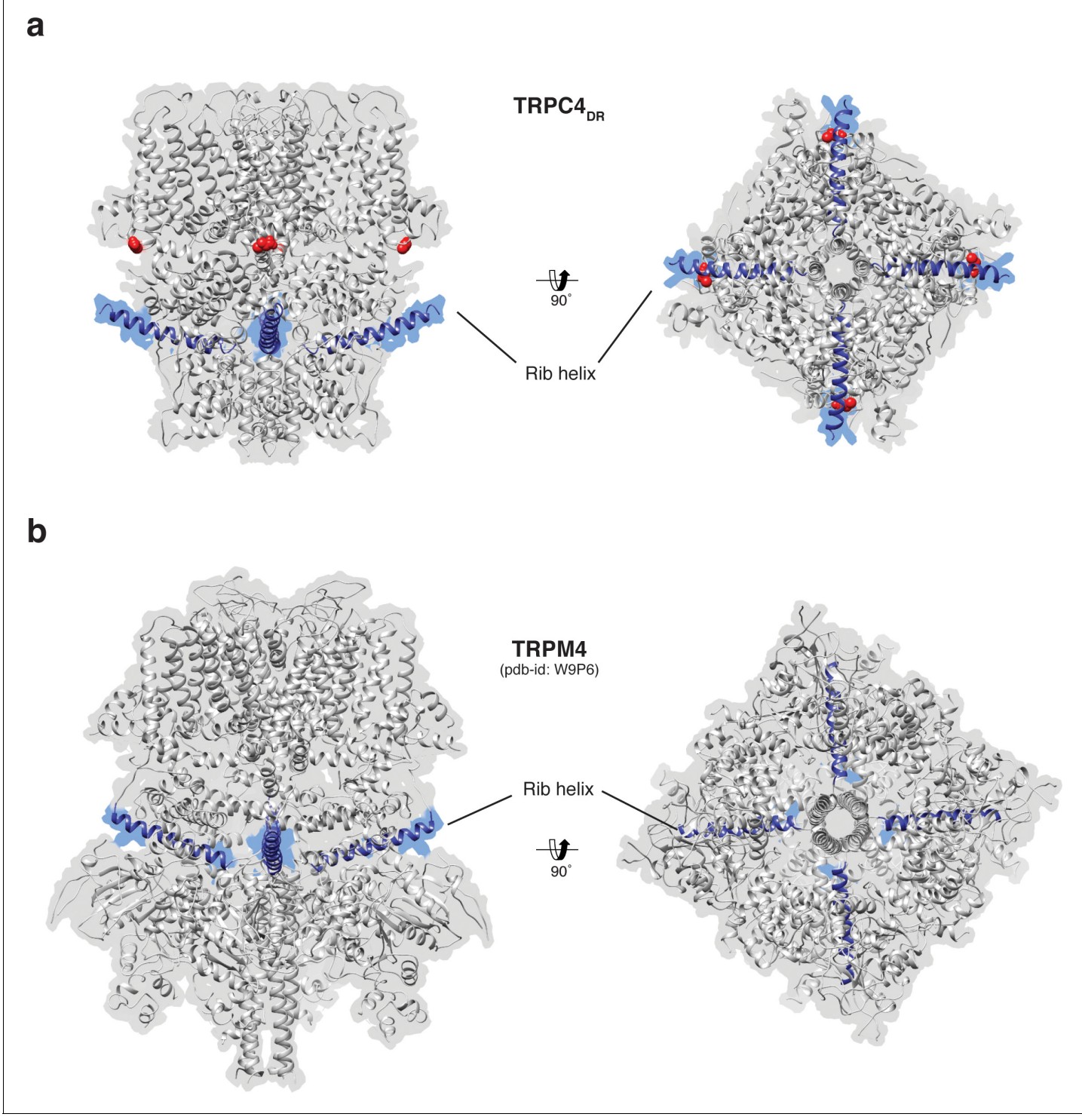

**Figure 7.** Position of the Rib helix in TRPC4$_{DR}$ and TRPM4. (a-b) Side and bottom view of the atomic models of TRPC4$_{DR}$ (a), and TRPM4 (b). The Rib helix is highlighted in blue color and the density of the map is shown in the background with high transparency. E648 in TRPC4$_{DR}$ is shown in sphere representation and colored red.

DOI: https://doi.org/10.7554/eLife.36615.018

The following figure supplement is available for figure 7:

**Figure supplement 1.** Predicted model of TRPC4$_{DR}$ interaction with IP$_3$ receptors and calmodulin.

DOI: https://doi.org/10.7554/eLife.36615.019

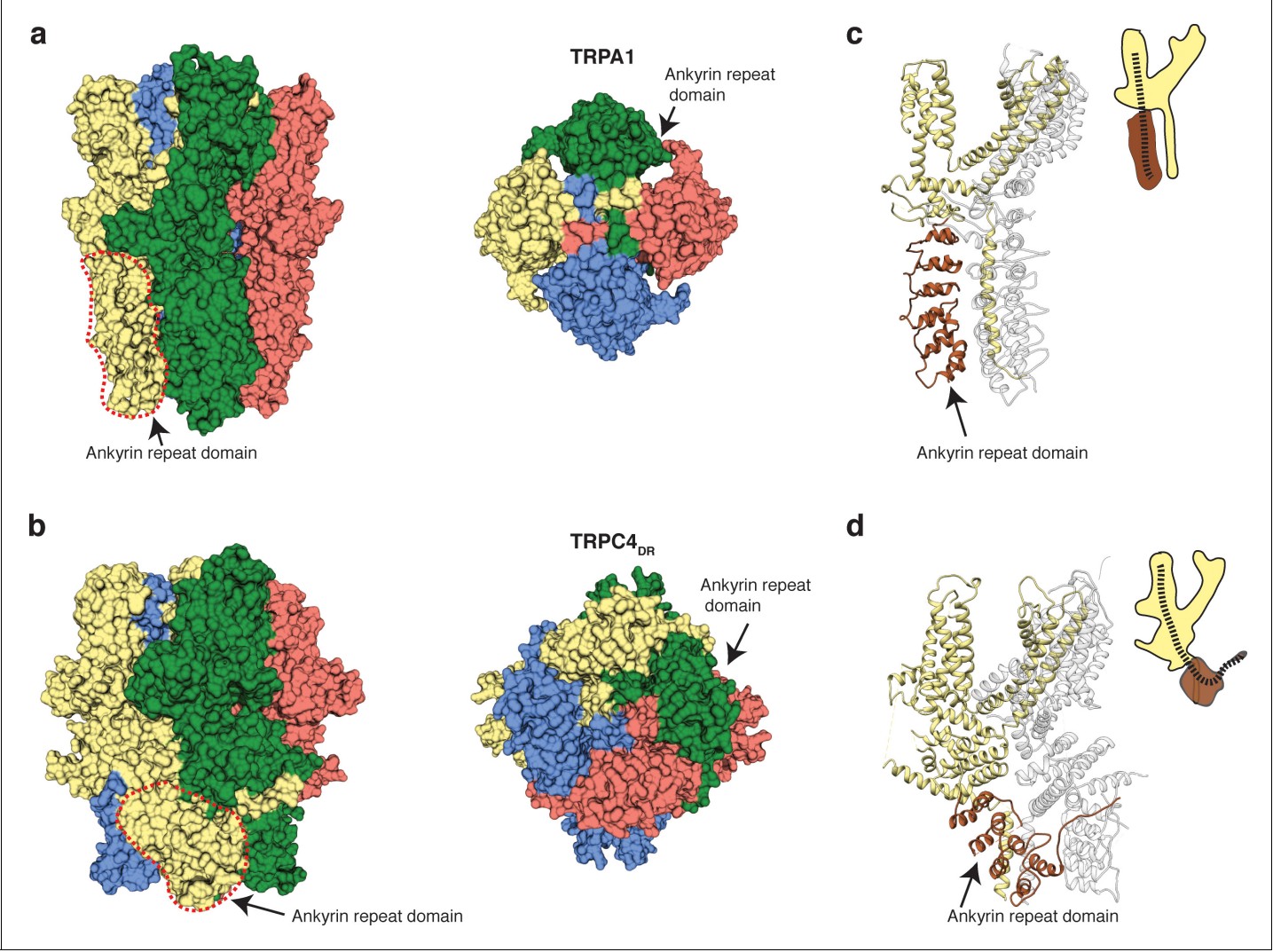

**Figure 8.** Comparison of the ankyrin domain arrangement of TRPC4$_{DR}$ and TRPA1. (a-b) Side and bottom view of the structures of TRPA1 (PDB-ID: 3J9P) (a) and TRPC4$_{DR}$ (b) in surface representation. Each subunit is colored with unique colors. (c–d) Ribbon and cartoon representation of TRPA1 (c) and TRPC4$_{DR}$ (d) showing two protomers in side view. The ankyrin repeats of one protomer are highlighted in brown.
DOI: https://doi.org/10.7554/eLife.36615.020

interaction has not been observed in the structures of other TRP subfamily members, such as TRPA1 (*Paulsen et al., 2015*), NOMPC (*Jin et al., 2017*) and TRPM4 (*Autzen et al., 2018*; *Guo et al., 2017a*; *Winkler et al., 2017*). In the TRPV subfamily, the cytoplasmic inter-protomer interaction is mediated between the ankyrin repeat domain of one protomer and the β-strand linker domain of the adjacent protomer (*Liao et al., 2013*).

## The lower part of the cytoplasmic domain

Ankyrin repeats, the Rib helix, and C-terminal helix domain constitute the most distant region of TRPC4$_{DR}$ (*Figure 2c–d*). Ankyrin repeats, that are often involved in protein-protein interactions (*Sedgwick and Smerdon, 1999*), are abundant in TRP channels in varying numbers, ranging from 0 in TRPP (*Shen et al., 2016b*) and TRPML (*Chen et al., 2017a*; *Hirschi et al., 2017*; *Schmiege et al., 2017*; *Zhou et al., 2017*) to 29 in NOMPC (*Jin et al., 2017*). In most structures of TRPs the first ankyrin repeats are oriented in a parallel fashion like in TRPA1 (*Paulsen et al., 2015*) (*Figure 8a*). TRPCs contain four ankyrin repeats in the cytoplasmic domain. Biochemical studies have shown that at least the first ankyrin repeat is necessary in TRPC5 for proper tetramerization and function of the

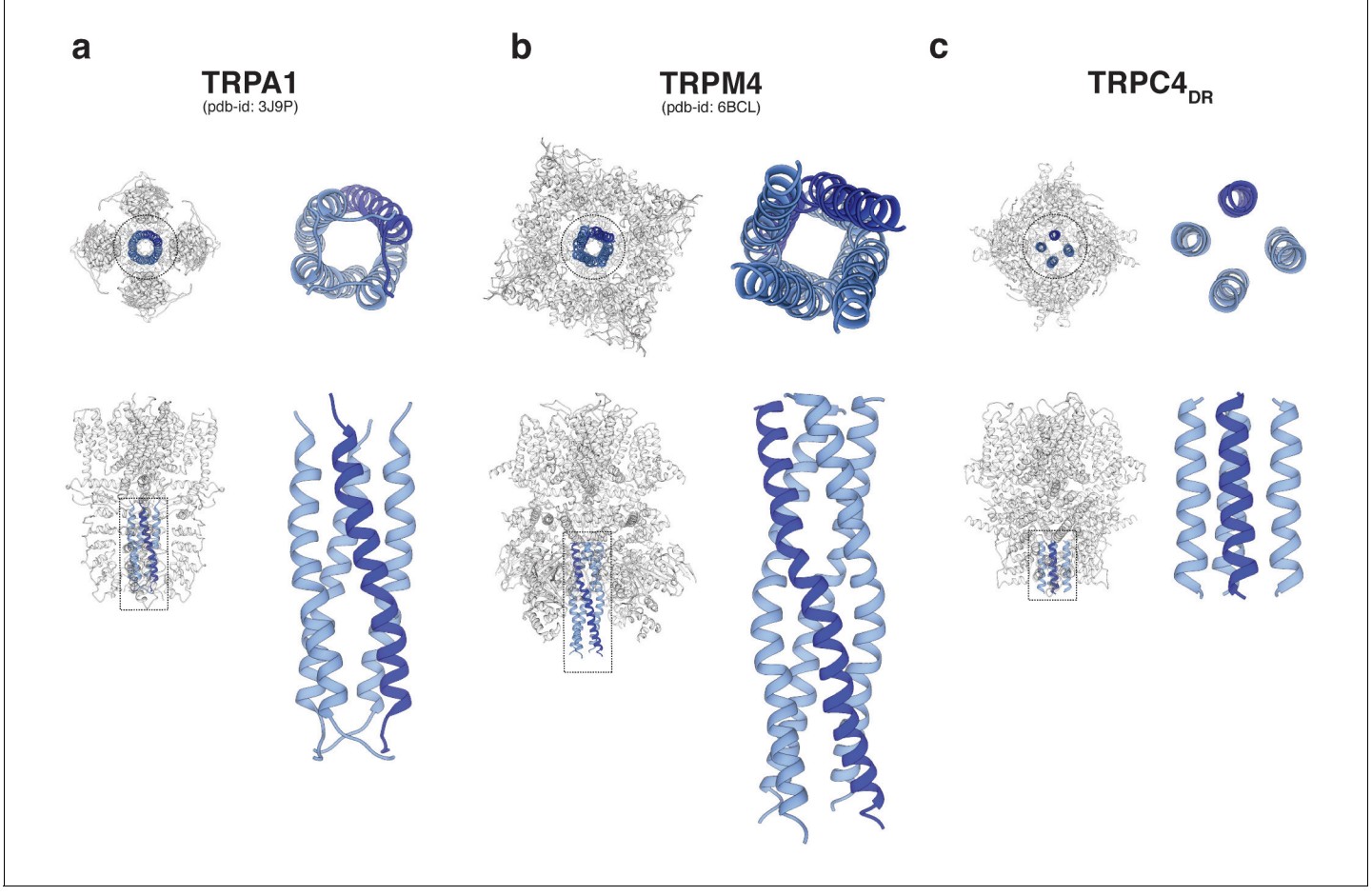

**Figure 9.** Comparison of the C-terminal helix architecture in TRPA1, TRPM4 and TRPC4_DR. (a-c) Each panel shows the complete tetramer in bottom and side view on the left and the zoomed-in view of the C-terminal helix alone in bottom and side view on the right. The C-terminal helix is shown in blue with one helix highlighted in shaded dark blue.

DOI: https://doi.org/10.7554/eLife.36615.021

channel (*Schindl et al., 2008*). In the structure of TRPC4_DR the repeats take a unique twisted orientation and swap over to an adjacent protomer interacting with the C-terminal helix domain and Rib domain explaining the ankyrin repeat's stabilizing effect on the tetramer (*Figures 2b* and *8b*).

The Rib helix runs almost parallel to the membrane and protrudes from the cytoplasmic domain (*Figure 7a*). It contains a dual calmodulin- and $IP_3$ receptor-binding site (CIRB) (*Mery et al., 2001*; *Tang et al., 2001*) and has been described as binding hub not only for these proteins but also for SESTD1 (*Miehe et al., 2010*) and the G protein $G_{\alpha i2}$ (*Jeon et al., 2012*) (*Zhu, 2005*). It is therefore a central interaction site for TRPC4-modulating proteins and the activity of the channel might be modulated by a displacement mechanism in which the different modulators compete for the same binding site (*Zhu, 2005*). A homologous helix has been observed in the structure of TRPM4 (*Autzen et al., 2018*; *Guo et al., 2017a*; *Winkler et al., 2017*). However, in the structure of TRPM4 the Rib helix is buried by other domains and only the very tip is accessible from the surface (*Figure 7b*). The Rib helix of TRPC4_DR, however, is accessible from the cytoplasm and direct binding of calmodulin or $IP_3$-receptor is possible (*Figure 7—figure supplement 1*).

A second major protein interaction hub is the C-terminal helix domain. It has only been observed before in TRPA1 (*Paulsen et al., 2015*), TRPM4 (*Autzen et al., 2018*) (*Guo et al., 2017b*; *Winkler et al., 2017*), and TRPM8 (*Yin et al., 2018*). In all described structures, it has a coiled-coil structure (*Figure 9a–b*). In the case of TRPC4_DR, however, the helices run in a parallel fashion (*Figure 9c*). The C-terminal helix domain has been shown to bind to the $D_2$ dopamine receptor

(*Hannan et al., 2008*) and spectrin αII and βV which are involved in surface expression and activation of TRPC4 (*Odell et al., 2008*).

Our structure of TRPC4 misses 162 residues at the C-terminus. This part of the protein comprises a PDZ-binding domain (*Mery et al., 2002*) and a second $IP_3$-binding domain (*Mery et al., 2001*) and calmodulin binding sites (*Tang et al., 2001*; *Trost et al., 2001*). Interestingly, the largest part of this region is missing in the second most abundant splice variant of TRPC4, namely TRPC4β (*Freichel et al., 2014*). In line with our structural data, a sequence-based structure prediction using PSIPRED (*Buchan et al., 2013*) indicates that this region has no defined secondary structure.

In summary, the results allow us to understand the three-dimensional organization of TRPC channels. The structure of $TRPC4_{DR}$ reveals how the ion conduction pathway is built and our results point towards a gating mechanism that is conserved in all type I TRP channels. Future studies will be needed to structurally understand how TRPCs are activated or in general modulated by proteins and other factors and permeate cations. Our structure-based localization of previously biochemically identified protein binding sites provides the structural framework for further experiments towards understanding how binding of regulatory protein affect the structure and function of TRPC4.

### Data availability

The atomic coordinates and cryo-EM maps for $TRPC4_{DR}$ are available at the Protein Data Bank (PDB)/Electron Microscopy Data Bank (EMDB) databases. The accession numbers are 6G1K/EMD-4339. The data sets generated in the current study are available from the corresponding author on reasonable request.

## Materials and methods

**Key resources table**

| Reagent type (species) or resource | Designation | Source or reference | Identifiers | Additional information |
|---|---|---|---|---|
| Gene (Danio rerio) | $TRPC4_{DR}$ | N/A | NCBI Reference sequence: NM_001289881 | Genes ordered from GenScript |
| Cell line (HEK293 GnTI-) | HEK293 GnTI- | ATCC | CRL-3022 | |
| Cell line (HEK293) | HEK293 | ATCC | CRL-1573 | |
| Cell line (Sf9) | Sf9 | N/A | Cat.No.600100 | Purchased from Oxford Expression Technologies Ltd (UK) |
| Recombinant DNA reagent | pcDNA3.1+$TRPC4_{DR}$ (See Materials and methods section for details) | This paper | | Cloned by GenScript in the customized pcDNA3.1 vector harbouring $N_{term}$-HIS$_8$tag-TEV- and $C_{term}$-HRV3C-eGFP |
| Recombinant DNA reagent | pEG BacMam | Eric Gouaux Lab PMID: 25299155 | | |
| Recombinant DNA reagent | pEG BacMam +$TRPC4_{DR}$ (See Materials and methods section for details) | This Paper | | Subcloned the pcDNA construct harbouring His-TEV- TRPC4- HRV3C -eGFP into the pEG BacMam with the introduction of StrepII tag. |
| Chemical compound, drug | (-)Englerin A | N/A | | Obtained from Lead Discovery Center GmbH, Dortmund Germany |
| Software, algorithm | SPHIRE software package | *Moriya et al., 2017* PMID: 28570515 | | |
| Software, algorithm | CrYOLO | N/A | | Wagner et al., unpublished. manuscript in preparation |
| Software, algorithm | RosettaCM | *Wang et al. (2015)* PMID: 27572730 | | |
| Software, algorithm | Rosetta | *Wang et al. (2015)* PMID: 27572730 | | |
| Software, algorithm | rosettaES | *Frenz et al. (2017)* PMID: 28628127 | | |
| Software, algorithm | Chimera | *Pettersen et al. (2004)* PMID:15264254 | | |

## Protein expression and purification

The full-length (2-915) *Danio rerio* (zebrafish) TRPC4$_{DR}$ (NM_001289881) was cloned into the pEG BacMam vector (*Goehring et al., 2014*), with a C-terminal HRV-3C cleavage site followed by EGFP Twin-StrepII-tag (WSHPQFEKGGGSGGGSGGSAWSHPQFEK) and a N-terminal eight poly-histidine tag followed by a TEV cleavage site. Bacmid and baculovirus were produced as described previously (*Goehring et al., 2014*). In brief, P2 baculovirus produced in Sf9 cells, was added to HEK293 GnTI⁻ cells (mycoplasma test negative, ATCC #CRL-3022) grown in suspension in FreeStyle medium (GIBCO-Life Technologies #12338–018) supplemented with 2% FBS at 37°C and 8% CO$_2$. Eight hours after transduction, 5 mM sodium butyrate was added to enhance protein expression for additional forty hours and the temperature was reduced to 30°C. After 48-hr post-transduction, cells were harvested by low-speed centrifugation in an Avanti J-20 XP centrifuge (Beckman Coulter) at 8,983 g for 15 min, washed in phosphate-buffered saline (PBS) pH 7.4, and pelleted in an Allegra X-15R (Beckman Coulter) Centrifuge at 4,713 g for 15 min. The cell pellet was resuspended and cells were lysed in an ice-cooled Microfluidizer Mod. 110S (Microfluidics Corporation) in buffer A (PBS buffer pH 7.4, 1 mM Tris(2-carboxyethyl)phosphine (TCEP), 10% glycerol) and protease inhibitors (0.2 mM AEBSF, 0.1 μM aprotonin and 1 μM phosphoramidion); 50 ml was used per pellet obtained from 800 ml of HEK 293 cell culture. Subsequently, the lysate was centrifuged at 15,000 g for 5 min, and the membranes were collected by ultracentrifugation in an Optima XPN-80 ultracentrifuge (Beckman Coulter) equipped with a Type 70 Ti Rotor at 164,700 g for one hour. The membranes were then mechanically homogenized in buffer B (100 mM Tris-HCl pH 8, 150 mM NaCl, 1 mM TCEP, 10% glycerol) and protease inhibitors, quick-frozen and stored at −80°C till further purification. Membranes were solubilized for 2 hr in buffer B supplemented with 1% DDM/0.1% CHS (Anatrace #D310-CH210). Insoluble material was removed by ultracentrifugation for 1 hr in a Beckman Coulter Type 70 Ti Rotor at 164,700 g. The soluble membrane fraction was diluted 2-fold with buffer B to reduce the detergent concentration and slowly applied to a column packed with Strep-Tactin beads (IBA Lifesciences) by gravity flow (6–10 s/drop) at 4°C. Next, the resin was washed with five column volumes of buffer B supplemented with 0.04% DDM/0.004% CHS solution, 0.02 mg ml$^{-1}$ soy polar lipids (Avanti #541602) dissolved in DDM and protease inhibitors. Bound protein was eluted seven times with 0.5 column volumes of buffer A with 4 mM d-desthiobiotin (Sigma), 0.026% DDM/0.0026% CHS, 0.02 mg ml$^{-1}$ soy polar lipids and 0.1 mM AEBSF protease inhibitor. The C-terminal EGFP tag was removed by incubating the eluted fractions with HRV-3C protease overnight. The next day, the detergent was replaced with amphipols A8-35 (Anatrace #A835) by adding four times the total protein mass and incubating for 6 hr at 4°C. Detergent removal was performed by adding Bio-beads SM2 (BioRad # 1523920) pre-equilibrated in PBS to the protein solution at 15 mg ml$^{-1}$ final concentration for 1 hr, then replaced with fresh Biobeads at 20 mg ml$^{-1}$ for overnight incubation at 4°C. Biobeads were removed using a Poly-Prep column (BioRad #7311550) and the solution was centrifuged at 20,000 g for 10 min to remove any precipitate. The protein was concentrated with a 100 MWCO Amicon centrifugal filter unit (Millipore) and purified by size exclusion chromatography using a Superose 6 10/300 GL column (GE healthcare) equilibrated in buffer C (PBS pH 7.4, 1 mM TCEP). The peak corresponding to tetrameric TRPC4$_{DR}$ in amphipols was collected and concentrated up to 0.3 mg ml$^{-1}$ for both negative stain and cryo-EM analysis.

## EM data acquisition

Tetramer TRPC4$_{DR}$ integrity was evaluated by negative stain electron microscopy prior to cryo-EM grid preparation and image acquisition (*Gatsogiannis et al., 2016*). In brief, 4 μl of TRPC4$_{DR}$ in amphipols at a concentration of 0.02 mg ml$^{-1}$ were applied onto a freshly glow-discharged copper grid (Agar Scientific; G400C) with an additional thin carbon layer. After an incubation of 45 s, the sample was blotted with Whatman no. four filter paper and stained with 0.75% uranyl formate. The images were recorded manually with a JEOL JEM-1400 TEM, operated at an acceleration voltage of 120 kV, and a 4,000 × 4,000 CMOS detector F416 (TVIPS) with a pixel size of 1.84 Å/pixel (*Figure 2—figure supplement 1c–e*). For cryo-EM, 3.5 μl of TRPC4$_{DR}$ at a concentration of 0.25 mg ml$^{-1}$ were applied onto freshly glow-discharged holey carbon grids (Quantifoil grid (1.2/1.3) 300 mesh) blotted using 2.5 s blotting time, 1 s draining time, 0 blotting force with 100% humidity at 4°C and vitrified in liquid ethane cooled by liquid nitrogen using a Vitrobot Mark III (FEI Company). The quality of the grids was screened with a JEOL JEM 3200 FSC electron microscope equipped with a

field-emission gun and an in-column energy filter operated at an acceleration voltage of 200 kV. The grids were then stored in liquid nitrogen.

## Electron microscopy and single particle cryo-EM data processing

A cryo-EM data set of TRPC4$_{DR}$ in amphipols was collected on a C$_s$-corrected TITAN KRIOS electron microscope (FEI), equipped with a high-brightness field-emission gun (XFEG) operated at an acceleration voltage of 300 kV. The images were acquired on a K2 summit direct electron detector (Gatan) operated in counting mode with a calibrated pixel size of 1.09 Å/pixel on the sample level with a post column GIF BioQuantum LS energy filter (Gatan) using a slit width of 20 eV. Two data sets were collected. The first data set with a total of 2020 images was collected with sixty frames (200 ms/frame) and an exposure of 12 s with a total dose of ~69.0 e$^-$ Å$^{-2}$. A second data set with a total of 1870 images was recorded with forty frames (300 ms/frame) and an exposure of 12 s with a total dose of ~74.4 e$^-$ Å$^{-2}$. All images were collected automatically using EPU (FEI). Motion correction was performed using the MotionCor2 program (*Zheng et al., 2017*).

All image processing was performed with the SPHIRE software package (*Moriya et al., 2017*) (*Figure 2—figure supplement 3*). Motion-corrected sums without dose weighting were used to determine the defocus and astigmatism in CTER (*Moriya et al., 2017*). The defocus range of the selected images was 0.84–2.93 μm (*Table 1*). Images with low quality were removed using the graphical CTF assessment tool in SPHIRE (*Moriya et al., 2017*). Motion-corrected sums with dose weighting were used for all other image processing steps. 426,377 single particles were picked automatically using CrYOLO (Wagner et al., unpublished). The particles were windowed to a final box size of 224 × 224 pixels. Reference-free 2-D classification and cleaning of the data set was performed with the iterative stable alignment and clustering approach ISAC (*Yang et al., 2012*) in SPHIRE. ISAC was performed at a pixel size of 3.02 Å/pixel. The 'Beautify' tool of SPHIRE was then applied to obtain refined and sharpened 2-D class averages at the original pixel size, showing high-resolution features (*Figure 2—figure supplement 2a*). A subset of ~132,622 particles producing 2-D class averages and reconstructions with high-resolution features were then selected for further structure refinement. An initial model for the first 3-D refinement was generated from the ISAC class averages with RVIPER. All 3-D refinements and classifications were performed imposing C4 symmetry. The 'clean' data set after ISAC was then subjected to 3-D refinements in MERIDIEN with a mask including amphiphols (*Moriya et al., 2017*). The final half-maps were combined, a tight mask and a B factor of $-130.0$ Å$^2$ were applied using SPHIRE's PostRefiner tool. This resulted in a cryo-EM map with an average resolution of 3.6 Å, as estimated by the 'gold standard' FSC = 0.143 criterion between the two masked half-maps (*Figure 2—figure supplement 2f*). The estimated accuracy of rotation and translation during the last iteration of the 3-D refinement were estimated to be 0.9375° and 0.7 pixels, respectively. Local FSC calculation was performed using the 'Local Resolution' tool in SPHIRE. This analysis showed that the core of TRPC4$_{DR}$ was resolved up to 3.1 Å resolution, whereas the upper and peripheral part of the protein showed the lowest resolution (~4.5–5 Å) (*Figure 2—figure supplement 2e*). The final density was then locally filtered on the basis of the local resolution with the 'LocalFilter' utility in the SPHIRE software package. Details related to data processing are summarized in *Table 1*.

3-D clustering into six groups was performed using the RSORT3D tool of SPHIRE with a 3-D focused binary mask including the C-terminal helix and the ankyrin repeats. The resulting volumes were refined with the 'local refinement mode' of MERIDIEN in SPHIRE. The SPHIRE 'PostRefiner' tool was used to determine the resolution of the locally refined volumes.

## Model building, refinement and validation

Initially, we built a homology model of TRPC4$_{DR}$ with Modeller (*Eswar et al., 2008*), using the structures of NOMPC (PDB-ID: 5VKQ) and TRPV1(PDB-ID: 3J5Q) as templates. These are the homologs with the highest sequence identity and of which high-resolution structures have been determined. Since only the transmembrane domain could be built using this method we used Rosetta to complete the model. Gaps in the membrane domain were built using RosettaCM (*Wang et al., 2015*). The rest of the monomer was built de novo using several iterative runs of the fragment fitting protocol implemented in Rosetta (*Wang et al., 2015*). After this, loops still missing from the model were manually built in Coot (*Emsley et al., 2010*). We generated a tetramer model by fitting four copies

of the monomer model into the cryo-EM density using Chimera (*Pettersen et al., 2004*). With the tetramer, we built 18 additional amino acids the N-terminal end of the protein using the enumerative sampling strategy in Rosetta (rosettaES) (*Frenz et al., 2017*), which could not be reliably built manually due to the quality of the density in this region. Final manual model building, including fitting of lipids, was done in Coot (*Emsley et al., 2010*). The full model was finally refined in Phenix (*Adams et al., 2010*) using the real space refinement protocol. The final model comprises residues 18–753 with missing sequence in between 119 and 134, 173 and 186, 273 and 283, 320 and 323, 389, 390, 649 and 651, 661 and 695, 728 and 730. Finally, we used Molprobity (*Chen et al., 2010*) to validate the overall geometry of the model, Phenix (*Adams et al., 2010*) to calculate the model-to-map correlation, and EMRinger (*Barad et al., 2015*) to validate the side chain geometry. The densities corresponding to annular lipids were modelled as phosphatidic acid lipid with a shorter lipid tail (PDB-ID 44E) and CHS (PDB-ID YO1). The geometric restraints for the refinement of both lipids were obtained by the eLBOW module in PHENIX (*Adams et al., 2010*). Figures were prepared in Chimera (*Pettersen et al., 2004*) and VMD (*Humphrey et al., 1996*). Structure-based sequence alignment was done using the Multiseq plugin (*Roberts et al., 2006*) in VMD. Multiple sequence alignment was done using Clustal Omega (*Sievers et al., 2011*). Figures of the sequence alignment were made with Jalview (*Waterhouse et al., 2009*). The radius of the pore was determined using HOLE (*Smart et al., 1996*). The $pK_a$ values of protein residues were calculated using the H++ server (*Anandakrishnan et al., 2012*) and used to assign the right protonation states in the calculation of electrostatic potentials in Chimera.

## Electrophysiological recordings on HEK293 cells heterologously expressing TRPC4$_{DR}$-EGFP

HEK293 cells (ATCC, CRL-1573$^{TM}$, Manassas, USA) were cultured at 37°C and 5% $CO_2$ in DMEM (Sigma, St. Louis, USA) supplemented with 10% fetal calf serum (Sigma, St. Louis, USA), and 5% penicillin/streptomycin (Sigma, St. Louis, USA). One day prior to transient transfections the HEK293 cells were seeded on 24-well plates. The transient transfections of the seeded HEK293 cells with pCDNA3.1 carrying TRPC4$_{DR}$-EGFP using Lipofectamine 2000 (Invitrogen, Carlsbad, USA) were performed two days prior to the patch-clamp measurements.

Whole cell patch-clamp experiments on HEK293 cells heterologously expressing TRPC4$_{DR}$-EGFP were performed under voltage clamp conditions using the Axopatch 200B amplifier (Axon Instruments, Union City, USA) and the DigiData 1322A interface (Axon Instruments, Union City, USA). Patch pipettes with resistances of 2–5 mΩ were fabricated from thin-walled borosilicate glass on a horizontal puller (Model P-1000, Sutter Instruments, Novato, USA). The series resistance was <10 MΩ. The bath solution (*Akbulut et al., 2015*) contained 135 mM NaCl, 5 mM KCl, 1.2 mM $MgCl_2$, 1.5 mM $CaCl_2$, 8 mM glucose, 10 mM HEPES (pH titrated to 7.4 using NaOH) and the pipette solution contained 145 mM CsCl, 2 mM $MgCl_2$, 10 mM HEPES, 1 mM EGTA (sodium salt), 0.1 mM GTP (sodium salt, pH titrated to 7.2 with CsOH). For the recordings of the IV-curves the membrane potentials were clamped to values ranging from −90 mV to +90 mV. The measurements were conducted in the absence and in the presence of 50 nM (-)-Englerin A.

## Acknowledgements

We thank Peter Nussbaumer (Lead Discovery Center) for providing (-)-Englerin A and Eric Gouaux for the kind gift of the pEG BacMam vector. We thank the SPHIRE developer team, in particular Pawel A Penczek for developing the cryo-EM image processing software used in this study. We are thankful to Nina Ludwigs and Marion Hülseweh for technical assistance. This work was supported by the Max Planck Society (to SR), and the European Council under the European Union's Seventh Framework Programme (FP7/2007–2013) (grant no. 615984) (to SR).

## Additional information

### Funding

| Funder | Grant reference number | Author |
|---|---|---|
| Max-Planck-Gesellschaft | Open-access funding | Stefan Raunser |
| European Commission | 615984 | Stefan Raunser |

The funders had no role in study design, data collection and interpretation, or the decision to submit the work for publication.

### Author contributions

Deivanayagabarathy Vinayagam, Data curation, Formal analysis, Validation, Investigation, Visualization, Writing—review and editing; Thomas Mager, Data curation, Formal analysis, Validation, Visualization, Writing—review and editing; Amir Apelbaum, Arne Bothe, Methodology, Writing—review and editing; Felipe Merino, Formal analysis, Validation, Writing—review and editing; Oliver Hofnagel, Data curation, Writing—review and editing; Christos Gatsogiannis, Formal analysis, Supervision, Validation, Visualization, Writing—review and editing; Stefan Raunser, Conceptualization, Formal analysis, Supervision, Funding acquisition, Investigation, Writing—original draft, Project administration, Writing—review and editing

### Author ORCIDs

Felipe Merino (iD) http://orcid.org/0000-0003-4166-8747
Stefan Raunser (iD) http://orcid.org/0000-0001-9373-3016

### Decision letter and Author response

Decision letter https://doi.org/10.7554/eLife.36615.028
Author response https://doi.org/10.7554/eLife.36615.029

## Additional files

### Supplementary files

• Transparent reporting form
DOI: https://doi.org/10.7554/eLife.36615.022

### Data availability

The atomic coordinates and cryo-EM maps for TRPC4DR are available at the Protein Data Bank (PDB)/Electron Microscopy Data Bank (EMDB) databases. The accession numbers are 6G1K/EMD-4339. The raw data sets generated in the current study (motion corrected and dose weighted mrc files and box files; 215 GB) are available from the corresponding author upon request.

The following datasets were generated:

| Author(s) | Year | Dataset title | Dataset URL | Database, license, and accessibility information |
|---|---|---|---|---|
| Vinayagam D, Mager T, Apelbaum A, Bothe A, Merino F, Hofnagel O, Gatsogiannis G, Raunser S | 2018 | Electron cryo-microscopy structure of the canonical TRPC4 ion channel | https://www.rcsb.org/structure/6G1K | Publicly available at the RCSB Protein Data Bank (accession no. 6G1K) |
| Vinayagam D, Mager T, Apelbaum A, Bothe A, Merino F, Hofnagel O, Gatsogiannis G, Raunser S | 2018 | Electron cryo-microscopy structure of the canonical TRPC4 ion channel | http://www.ebi.ac.uk/pdbe/entry/emdb/EMD-4339 | Publicly available at the EMDataBank (accession no. EMD-4339) |

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
