## [Decision Letter]

Thank you for submitting your article "Electron cryo-microscopy structure of the canonical TRPC4 ion channel" for consideration by *eLife*. Your article has been reviewed by three peer reviewers, and the evaluation has been overseen by Kenton Swartz as the Reviewing Editor and Richard Aldrich as the Senior Editor. The following individual involved in review of your submission has agreed to reveal his identity: László Csanády (Reviewer #2).

The reviewers have discussed the reviews with one another and the Reviewing Editor has drafted this decision to help you prepare a revised submission.

Summary:

The TRP family of cation channels is comprised of 7 subfamilies with varying numbers of members, all of which exhibit different functional properties, ion selectivity and modulation by various interaction partners and cellular mediators. They are thus involved in a large variety of physiological functions, from sensory stimuli detection to ion homoeostasis in the body, to proper organelle function inside the cell. Mutations in many of these channels have also been directly associated with serious human pathologies. The recent cryo-EM revolution has brought to light the structures of many TRP channels, providing insight into their mechanisms of gating and modulation, and highlighting both the striking similarities and differences between members of the TRP family. So far structures of representatives from each of the TRP channel subfamilies have been obtained except for the TRPC family. TRPC (canonical) channels are widely expressed in the body. The TRPC4 and C5 have been shown to form non-selective Ca^2+^ permeable channels activated upon Ca^2+^-store depletion, and their function is modulated by interaction with important partners such as calmodulin, IP3 receptors and STIM1.

In the present manuscript the authors have solved the first structure of a TRPC channel, the zebra fish TRPC4 in the closed apo state, at an overall resolution of 3.6A using cryo-EM with amphipol-stabilized proteins expressed in HEK cells prior to purification. The map quality allowed for the de novo building of a structural model comprising 70% of the protein. The authors first demonstrate that the zebrafish TRPC4 is functional and can be activated by the agonist (-)-EnglerinA and exhibits similar rectification properties as human TRPC4. Their structure reveals important properties of the channel, with interesting similarities and differences with other TRP channels which allow for inferences on the function and modulation of TRPC channels and TRP channels in general. The tetrameric protein exhibits a domain-swapped architecture for the transmembrane region. The transmembrane domain topology is very similar to the rest of the TRP channels which lack large extracellular domains (i.e. all except TRPP1 and TRPMLs). The structure allows visualization of the ion conduction pathway, revealing interesting residue interactions with functional significance in the outer vestibule, the pore helix and S6 segments, and also demonstrate the existence of an intracellular activation gate within the S6 helix.

Other functionally important interactions between the TRP domain and the M2 motif are also revealed, together with interesting differences between the voltage sensor-like domain of TRPC4 and other related TRP channels such as TRPV1 and M8. Two lipid binding sites are identified, one within the VSL domain/linker domain helix and the other within the upper half of the pore. Interestingly, the authors find that a functionally important Ca^2+^ binding site within the VSL-domain identified in the Ca^2+^-bound TRPM4 structure by Autzen et al. is conserved in TRPC4, suggesting that the Ca^2+^-dependent modulation of this channel, for which no structural correlate had been found to date, might depend on this site. The intracellular domains of TRPC4 are unique and exhibit a two-layered architecture with extensive intra- and inter-subunit contacts that likely contribute to the stability of the tetramer and gating of the receptor. The presence of a protruding C-terminal rib helix known to contain interaction sites for several important TRPC4 modulators, including calmodulin, IP3R, SESTD1 and G proteins, is resolved and shown to be exposed to the cytosol, providing a structural background to investigate the complex regulatory mechanisms of this protein. Overall, the work in the manuscript seems of high quality, and the properties of the structure are discussed with detail, considering both functional observations in TRPC4 and the related TRPC5 channels and the structural information coming from the other TRP channels.

Essential revisions:

1) In Figure 1, the authors need to include current traces for the non-transfected cells in the absence and presence of agonist. They should also include a normalized mean +/- SEM current-voltage relations for both transfected and un-transfected cells in the absence and presence of the agonist.

2) The authors seem to imply that the disulfide bond formation/breakage in the extracellular pore loop of TRPM4 is Ca^2+^- and gating-dependent. However, disulfides are strong bonds that would not be expected to break simply from the binding of Ca^2+^ ions to a site very distant from the bond. Autzen et al. mention in their TRPM4 paper that the breakage of the bond in their Ca^2+^-bound structure could have resulted from radiation damage. It is thus necessary to modify that part of the Results/Discussion section in the present manuscript to reflect this. The disulfide formation/breakage might have a role in gating of the TRPC4 channel, but this most likely does not occur as a consequence of agonist binding, but rather by the direct action of oxidizing or reducing agents in the cysteines themselves.

3) The use of (-)-Englerin A to activate the channel in functional experiments naturally begs the question of whether a structure of activator and TRPC4 in complex could have reasonably been included to add mechanistic richness to the study. This is especially so given that reporting multiple states (when chemical tools exist) is typically straightforward with cryo-EM. If there were experimental challenges that precluded this, or if (-)-Englerin is not a good candidate for mechanistic insight, etc., perhaps the authors could comment on this.

4) The cryo-EM density map shows unambiguous extra density, which the authors interpreted as CHS and PA. Inspection of the map and model together (at a contour similar to the figures) shows portions of the CHS and PA models fall outside of the density. That said, CHS and PA are sensible candidates to assign to these densities, but given the 3.6 Å resolution of the map, did the authors try or consider trying an orthogonal method to firmly establish the identities of the lipids in the amphipol solubilized protein? For example, Mi and co-workers used mass spectrometry with mutagenesis to reinforce lipid density assignment for their MsbA work (Mi et al. 2017 Nature). We are not requesting new experiments, but it would be nice to know more if the authors have additional information or tried other approaches.

5) The function of voltage-sensing domains in voltage-activated channels depends not only on the presence of positively charged residues in the S4, but also on the presence of countercharges along the path through which the S4 moves upon activation/deactivation, together with a hydrophobic "plug" that focuses the electric field at a point between the intra-and extracellular cavities in which the S4 charges reside. The figures used to illustrate the properties of the VSL-domain of Kv, TRPC4, TRPM4 and TRPV1 channels focuses only on the S4 charges. It may be useful to inspect whether the other VSL-domain helices contain comparable properties as those of voltage-activated channels.

Furthermore, although it is an interesting observation that the S4 of TRPC4 is more hydrophilic than that of e.g. TRPV1, TRPM4 and TRPM8, the voltage-dependence of TRPC4 is rather weak, arguably weaker than that of TRPV1 and TRPM8. Yet, the latter channels are not expected to derive their voltage-dependence from the movement of S4. Thus, we suggest more caution in suggesting that the VSL-domain of TRPC4 might function similarly as that of voltage-gated channels. Finally, the TRPP1 channel seems to also contain a larger number of polar residues in its S4 helix, and exhibits pronounced voltage-dependence, such that it might be interesting to include it in the comparison.

---

## [Author Response]

Essential revisions:1) In Figure 1, the authors need to include current traces for the non-transfected cells in the absence and presence of agonist. They should also include a normalized mean +/- SEM current-voltage relations for both transfected and un-transfected cells in the absence and presence of the agonist.

We have added current traces for the non-transfected cells in the absence and presence of (-)-Englerin A to Figure 1. We also included normalized mean +/- SEM current-voltage relations for both transfected and un-transfected cells in the absence and presence of the agonist.

2) The authors seem to imply that the disulfide bond formation/breakage in the extracellular pore loop of TRPM4 is Ca^2+^- and gating-dependent. However, disulfides are strong bonds that would not be expected to break simply from the binding of Ca^2+^ ions to a site very distant from the bond. Autzen et al. mention in their TRPM4 paper that the breakage of the bond in their Ca^2+^-bound structure could have resulted from radiation damage. It is thus necessary to modify that part of the Results/Discussion section in the present manuscript to reflect this. The disulfide formation/breakage might have a role in gating of the TRPC4 channel, but this most likely does not occur as a consequence of agonist binding, but rather by the direct action of oxidizing or reducing agents in the cysteines themselves.

We apologize for creating confusion. We did not want to claim any Ca^2+^ dependence in connection to the disulphide. This passage was not well worded. We have rephrased this part of the manuscript to make it clearer for the reader.

3) The use of (-)-Englerin A to activate the channel in functional experiments naturally begs the question of whether a structure of activator and TRPC4 in complex could have reasonably been included to add mechanistic richness to the study. This is especially so given that reporting multiple states (when chemical tools exist) is typically straightforward with cryo-EM. If there were experimental challenges that precluded this, or if (-)-Englerin is not a good candidate for mechanistic insight, etc., perhaps the authors could comment on this.

Indeed, we have tried to obtain the structure of TRPC in complex with (-)-Englerin as suggested by the reviewers. However, although we have tried several conditions, so far we have not been successful. Since there is also no experimental evidence that (-)-Englerin binds directly to TRPC4 to activate it, we will now first investigate the interaction of (-)-Englerin with TRPC4 in an in vitro system. However, these studies will take time and go beyond the scope of this manuscript. We believe that our structure of TRPC4 in the apo state gives already enough novel insights to justify publication in *eLife*.

4) The cryo-EM density map shows unambiguous extra density, which the authors interpreted as CHS and PA. Inspection of the map and model together (at a contour similar to the figures) shows portions of the CHS and PA models fall outside of the density. That said, CHS and PA are sensible candidates to assign to these densities, but given the 3.6 Å resolution of the map, did the authors try or consider trying an orthogonal method to firmly establish the identities of the lipids in the amphipol solubilized protein? For example, Mi and co-workers used mass spectrometry with mutagenesis to reinforce lipid density assignment for their MsbA work (Mi et al. 2017 Nature). We are not requesting new experiments, but it would be nice to know more if the authors have additional information or tried other approaches.

Because of its distinguished flat shape one density clearly corresponds to sterol. Because we have included CHS in our purification and CHS has been identified at similar positions in the structures of TRPM4 and TRPML3, we have assigned CHS to it. Although protruding from the density there is enough space in the structure to accommodate the succinyl group of CHS which is likely stabilized by a putative hydrogen bond with Tyr315.

The second lipid density can be clearly assigned to phosphatidic acid (PA) or ceramide-1 phosphate. PA has also been assigned to a density found in the TRP channel polycytin-2. Because PA is a constituent of soy polar lipids extract and has therefore been included in our purification buffer we fitted PA into the density.

We explain now our approach for fitting the respective lipids in the revised manuscript.

5) The function of voltage-sensing domains in voltage-activated channels depends not only on the presence of positively charged residues in the S4, but also on the presence of countercharges along the path through which the S4 moves upon activation/deactivation, together with a hydrophobic "plug" that focuses the electric field at a point between the intra-and extracellular cavities in which the S4 charges reside. The figures used to illustrate the properties of the VSL-domain of Kv, TRPC4, TRPM4 and TRPV1 channels focuses only on the S4 charges. It may be useful to inspect whether the other VSL-domain helices contain comparable properties as those of voltage-activated channels.Furthermore, although it is an interesting observation that the S4 of TRPC4 is more hydrophilic than that of e.g. TRPV1, TRPM4 and TRPM8, the voltage-dependence of TRPC4 is rather weak, arguably weaker than that of TRPV1 and TRPM8. Yet, the latter channels are not expected to derive their voltage-dependence from the movement of S4. Thus, we suggest more caution in suggesting that the VSL-domain of TRPC4 might function similarly as that of voltage-gated channels. Finally, the TRPP1 channel seems to also contain a larger number of polar residues in its S4 helix, and exhibits pronounced voltage-dependence, such that it might be interesting to include it in the comparison.

As suggested by the reviewers we have now included also TRPM8 and TRPP1 in the comparison. We have also included an additional figure panel highlighting residues that provide countercharges to the basic residues in S4 and the hydrophobic patch. We have completely rephrased the paragraph describing the comparison between the VSL-domain of TRPC4 and that of voltage-gated ion channels. As suggested by the reviewers, we are now more careful in deriving a voltage-dependence from the movement of S4.